# Maternal and Intrauterine Influences on Feto-Placental Growth Are Accompanied by Sexually Dimorphic Changes in Placental Mitochondrial Respiration, and Metabolic Signalling Pathways

**DOI:** 10.3390/cells12050797

**Published:** 2023-03-03

**Authors:** Esteban Salazar-Petres, Daniela Pereira-Carvalho, Jorge Lopez-Tello, Amanda N. Sferruzzi-Perri

**Affiliations:** 1Departamento de Ciencias Básicas, Facultad de Ciencias, Universidad Santo Tomás, Valdivia 5090000, Chile; 2Centre for Trophoblast Research, Department of Physiology, Development and Neuroscience, University of Cambridge, Cambridge CB2 3EG, UK

**Keywords:** mitochondria, placenta, fetus, sex, PI3K

## Abstract

Adverse maternal environments such as small size, malnutrition, and metabolic conditions are known to influence fetal growth outcomes. Similarly, fetal growth and metabolic alterations may alter the intrauterine environment and affect all fetuses in multiple gestation/litter-bearing species. The placenta is the site of convergence between signals derived from the mother and the developing fetus/es. Its functions are fuelled by energy generated by mitochondrial oxidative phosphorylation (OXPHOS). The aim of this study was to delineate the role of an altered maternal and/or fetal/intrauterine environment in feto-placental growth and placental mitochondrial energetic capacity. To address this, in mice, we used disruptions of the gene encoding phosphoinositol 3-kinase (PI3K) p110α, a growth and metabolic regulator to perturb the maternal and/or fetal/intrauterine environment and study the impact on wildtype conceptuses. We found that feto-placental growth was modified by a perturbed maternal and intrauterine environment, and effects were most evident for wildtype males compared to females. However, placental mitochondrial complex I+II OXPHOS and total electron transport system (ETS) capacity were similarly reduced for both fetal sexes, yet reserve capacity was additionally decreased in males in response to the maternal and intrauterine perturbations. These were also sex-dependent differences in the placental abundance of mitochondrial-related proteins (e.g., citrate synthase and ETS complexes), and activity of growth/metabolic signalling pathways (AKT and MAPK) with maternal and intrauterine alterations. Our findings thus identify that the mother and the intrauterine environment provided by littermates modulate feto-placental growth, placental bioenergetics, and metabolic signalling in a manner dependent on fetal sex. This may have relevance for understanding the pathways leading to reduced fetal growth, particularly in the context of suboptimal maternal environments and multiple gestation/litter-bearing species.

## 1. Introduction

To meet fetal growth requirements, the placenta adapts and changes functionally under normal and altered intrauterine environments [1,2,3,4,5,6]. These changes are fundamental to maintain fetal growth trajectory and development and are aided by the endocrine output from the placenta that encourage physiological changes in the mother and ensure nutrients are available for transfer to the fetus [7,8,9,10]. From the perspective of the placenta, the intrauterine environment is a site of convergence between signals derived from the mother and those from the developing fetus or fetuses in the case of multiple gestation/litter-bearing species [3,11,12]. Mouse and human placentas are relatively similar in exchange, endocrine function, and morphology, as both are haemochorial in nature. However, the murine placenta is divided in two distinguished regions: the junctional zone (responsible for endocrine production) and the labyrinth zone (LZ; substrate exchange) [13,14,15], whereas in the human placenta, the villous syncytiotrophoblast performs both functions. Nevertheless, comparative proteomic and transcriptomic analyses of mouse placental LZ and human placental villous samples showed that over 80% of genes implicated in mouse placental phenotypes are co-expressed in both species [16]. Studies performed on mice and humans demonstrate that under unfavourable health conditions, maternal adaptations may fail, leading to abnormal nutrient partitioning and pregnancy complications. Under these adverse conditions, placental development may also be compromised, yet, in certain situations, the placenta may also be capable of maintaining or even increasing nutrient supply to optimize fetal growth in that prevailing environment [17,18,19,20,21,22,23]. However, further work is required to understand the role of maternal and fetal signals in conditioning the intrauterine environment and to uncover the cellular and molecular pathways that may modulate placental metabolic functions [24]. 

Adaptations in placental transport capacity could be related to changes in mitochondrial function. This is because the placenta relies heavily on energy (ATP) that is generated primarily by mitochondria. Mitochondria generate ATP via oxidative phosphorylation (OXPHOS) and reducing substrates derived from β-oxidation and the tricarboxylic acid cycle [25]. The energy generated by mitochondria is used for placental metabolism, growth, morphological remodelling, and to actively transport a range of substrates from the mother to the fetus for growth and development [26,27,28,29,30]. Prior work has reported that placental mitochondrial bioenergetic capacity alters developmentally to meet the increasing fetal demands for growth toward term in humans and animal models [31,32]. Moreover, there are changes in placental mitochondrial function in human pregnancies and rodents exposed to unfavourable maternal environments, which includes changes in OXPHOS and mitochondrial-related proteins [31,33,34,35,36,37,38,39]. Finally, accumulating findings are demonstrating that placental adaptive responses, and mitochondrial function specifically, are different for female and male fetuses [1,26,39,40,41]. However, it remains unclear how the maternal and intrauterine environments interact to modulate placental mitochondrial function and, consequently, how this interaction relates to the placental support of female and male fetal growth.

To address these key deficiencies in knowledge, this study used disruptions of the gene encoding phosphoinositol 3-kinase (PI3K) p110α (*Pik3ca*) in mice as a tool to perturb the intrauterine environment and assess the impact on feto-placental development, placental mitochondrial respiration, and metabolic signalling pathways of female and male wildtype fetuses. The PI3K signalling pathway is a critical signalling pathway that regulates cell metabolism and growth [42,43,44,45,46,47,48]. Moreover, this pathway mediates the metabolic effects of insulin [49,50,51] by promoting glucose uptake and glycogen synthesis [52,53]. This is highly relevant since previous work has found that PI3K-p110α signalling is required for mediating metabolic adaptations in the mother that support fetal glucose transfer during pregnancy [24,54,55]. Other work has also shown that PI3K-p110α signalling regulates placental LZ formation [21,56]. Finally, PI3K-p110α signalling deficiency in the fetus causes growth stunting and mal-formed placentas of both males and females, and the presence of PI3K-p110α mutants would be expected to influence the intrauterine environment and, hence, the development of wildtype siblings [21,24,56].

## 2. Materials and Methods 

### 2.1. Animals and Experimental Design

Mice were housed at the University of Cambridge Animal Facility and the study was undertaken abiding by the UK Home Office Animals (Scientific Procedures) Act 1986 after approval from the University of Cambridge ethics committee (UK HOL PP6324596). Mice were allowed to drink and feed ad libitum and were housed under a 12:12 h light-dark cycle. This study employed wildtype (WT) mice and mice with partial inactivation of the PI3K isoform p110α, which was induced via heterozygous inheritance of a dominant negative mutation in *Pik3ca* (*Pik3ca*-D933A; mice are defined as α/+ in the text). The generation of the α/+ mutant mice has previously been reported [49] and they have been on a C57BL/6 background for more than 10 generations. Female mice were virgin and aged 4-months when they were mated with males. Three types of crosses of mice were generated in this study (Figure 1). Briefly, WT females were mated with WT males (WT x WT) to generate pregnancies with control WT litters. WT females were mated with α/+ males (WT x α/+) to generate pregnancies with litters containing α/+ conceptuses (adverse intrauterine environment). Finally, α/+ females were mated with WT males (α/+ x WT) to generate pregnancies with litters containing α/+ conceptuses within a mother who was α/+ (adverse intrauterine and maternal environment) [24,54]. The detection of a mating plug in the female vagina was used to indicate gestational day 1 (Gd1).

### 2.2. Tissue Collection and Genotyping

Pregnant dams were killed by cervical dislocation on Gd18 for retrieval of the gravid uterus. Fetuses and placentas were dissected free of fetal membranes and individually weighed. After weighing the placenta, the placental labyrinth LZ was micro-dissected from the endocrine junctional zone. The LZ was then weighed and either immediately snap frozen or placed in cryopreservation media prior to snap freezing (0.21 M mannitol, 0.07 M sucrose, 30% DMSO, pH 7.5). These LZ samples were stored at −80 °C for subsequent molecular and mitochondrial respiratory analyses, respectively. Fetal tails were taken to determine sex (*Sry* gene F: 5′-GTGGGTTCCTGTCCCACTGC-3′, R: 5′-GGCCATGTCAAGCGCCCCAT-3′ and autosomal PCR control gene F: 5′-TGGTTGGCATTTTATCCCTAGAAC-3′, R: 5′-GCAACATGGCAACTGGAAACA-3′) and α/+ genotype (F: 5′-TTCAAGCACTGTTTCAGCT-3′ and R: 5′-TTATGTTCTTGCTCAAGTCCTA-3′) [55]. Only WT conceptuses were analysed in this study.

### 2.3. Placental LZ Mitochondrial Respirometry

Mitochondrial respiratory capacity was examined in placental LZ samples using high resolution respirometry (Oxygraph 2k respirometer; Oroboros Instruments, Innsbruck, Austria), using a sequential substrate, inhibitor, uncoupler titration (SUIT) protocol as reported previously [1]. In brief, thawed LZ samples (in sucrose solution, pH 7.5) were permeabilized using saponin (5 mg/ mL, Sigma-Aldrich, Gillingham, UK) in biopsy preservation medium (BIOPS; pH 7.1, containing 10 mM Ca-EGTA buffer, 0.1µM free Ca^2+^, 1 mM free Mg^2+^, 20 mM imidazole, 20 mM taurine, 50 mM K-MES, 0.5 mM DTT, 6.56 mM MgCl_2_, 5.77 mM ATP, and 15 mM phosphocreatine). Following washes in respiratory medium (MiR05; pH 7.1 solution containing 20 mM HEPES, 0.46 mM EGTA, 2.1 mM free Mg^2+^, 90 mM K^+^, 10 mM Pi, 20 mM taurine, 110 mM sucrose, 60 mM lactobionate, and 1 g/L BSA), 15–20 mg of LZ sample was analysed at 37 °C in a pre-calibrated Oxygraph-2k respirometer chamber. Oxygen concentration was kept between 250 μM and 300 μM, and real-time acquisition and assessment of oxygen consumption was obtained using the SUIT protocol [1] and DatLab software (V7, Oroboros Instruments). To provide information about outer membrane integrity, exogenous cytochrome c (10 µM) was added during maximum OXPHOS capacity (after succinate addition to activate CI and CII pathways in the presence of ADP), and LZ samples showing a >30% increase in oxygen consumption were excluded (a total of 6 placentas from different groups were excluded from the study), as suggested by others [57]. Respiration was expressed as oxygen consumption per mg of placental LZ tissue.

### 2.4. Western Blot Analysis

Protein abundance was determined in placental LZ samples (representative for each sex and genotype) using Western blotting as previously described [1]. Briefly, proteins were extracted using RIPA buffer (Thermo Fisher Scientific, Waltham, MA, USA) supplemented with Mini EDTA-free protease inhibitor cocktail mix (Roche, Basel, Switzerland, CH). Proteins were separated using electrophoresis and transferred onto 0.2 μm nitrocellulose membranes (Bio-Rad Laboratories, Hercules, CA, USA). Membranes were blocked in tris-buffered saline with Tween 20 (TBS-T) plus 5% milk or fetal bovine serum for one hour at room temperature and subsequently incubated overnight with primary antibodies described in Appendix A. The day after, membranes were washed in TBS-T and incubated with secondary antibodies (NA934 or NA931; 1:10,000). Protein bands were visualized via enhanced chemiluminescence using SuperSignal™ West Femto Substrate (Thermo Fisher Scientific, MA, USA). Signal intensity was determined using ImageJ, version 2.1.0 software, and proteins were normalized to protein loading as informed by Ponceau S staining [58].

### 2.5. Statistical Analysis

Statistical analyses were performed using GraphPad Prism version 9 (GraphPad, CA, USA). Outliers were detected by the Prisms Grubbs’ test. To analyse the effect of the mating cross (WT x WT, WT x α/+, and α/+ x WT) on WT conceptus development and LZ mitochondrial respirometry, each sex was analysed separately by one-way ANOVA followed by Tukey post hoc tests. Litter composition (sex and α/+ genotype) was analysed using Chi-squared analysis. Data are shown as mean ± SEM with individual datapoints. Data were considered statistically significant at values of *p* < 0.05. Any tendency for a significant effect (*p* < 0.07) are stated with specific *p* value in the text.

## 3. Results

### 3.1. Littermate and/or Maternal p110α Deficiency Alters Feto-Placental Growth of WT Fetuses in a Sex-Specific Manner

We used our three parental crosses (female x male: WT x WT, WT x α/+, and α/+ x WT, shown in Figure 1) to assess if littermate and/or maternal p110α deficiency affects the growth of WT conceptuses. In particular, by comparing to WT x WT, pregnancies generated by the WT x α/+ cross informed on the impact of adverse intrauterine conditions (presence of α/+ littermates in utero), whilst those created by mating α/+ females with WT mates informed on the combined effect of an adverse intrauterine and an adverse maternal environment [24,54]. Comparing WT x α/+ and α/+ x WT pregnancies allowed us to deduce the influence of an adverse maternal environment due to α/+ deficiency on WT conceptus growth. We found there was no effect of the parental cross on litter size or sex ratio and there was also no difference in the percentage of α/+ pups within the litter between WT x α/+ and α/+ x WT pregnancies (Appendix A). 

By analysing each fetal sex separately, we found that WT female fetuses from α/+ x WT pregnancies were growth-restricted when compared to those from either the WT x WT or WT x α/+ pregnancies (Figure 2A). Further, WT male fetuses from WT x α/+ pregnancies were significantly heavier when compared to those from the WT x WT and α/+ x WT pregnancies (Figure 2A). Placental weight of WT female fetuses did not vary between WT x WT, WT x α/+, and α/+ x WT pregnancies. In contrast, WT male placental weight was greater in α/+ x WT pregnancies compared to WT x WT and WT x α/+ pregnancies (Figure 2B). Similar to placental weight, no differences were detected in the LZ weight between groups for WT female fetuses, but LZ weight for WT male fetuses in α/+ x WT pregnancies was heavier when compared to WT x WT pregnancies (Figure 2C). Since our study is focused on responses of the placental LZ, which determines nutrient supply to the fetus for growth, we calculated the ratio between fetal weight and placental LZ weight. This calculation revealed that fetal weight as a proportion of LZ weight was not different among the groups in females but was lower in males from the α/+ x WT group when compared to WT x WT and WT x α/+ groups (Figure 2D). Collectively, these findings suggest that littermate and maternal p110α deficiency modifies WT conceptus growth. However, the specific nature of these changes depends on fetal sex.

### 3.2. Littermate and/or Maternal p110α Deficiency Alters Mitochondrial Bioenergetics in the Placental LZ of WT Fetuses of Both Sexes

To understand how littermate and maternal p110α deficiency modifies WT conceptus growth, we then assessed placental LZ mitochondrial function in WT fetuses from our three pregnancy groups (WT x WT, WT x α/+, and α/+ x WT). While there were no differences in Complex (C) CI_LEAK_ and CI_OXPHOS_ in both WT females and males between our different experimental groups (Figure 3A,B), maximal respiratory capacity through CI and II (CI + CII_OXPHOS_) was lower in the LZ of both female and male WT fetuses from WT x α/+ and α/+ x WT pregnancies compared to WT x WT pregnancies (Figure 3C). This was related to lower CII-associated oxygen consumption in WTs of both sexes in WT x α/+ and α/+ x WT versus WT x WT pregnancies (Figure 3D). It was also related to a reduction in oxygen consumption by the total ETS for female and male WTs in WT x α/+ and α/+ x WT litters (Figure 3E). Mitochondrial reserve capacity was profoundly decreased in the placental LZ of WT males from WT x α/+ and α/+ x WT pregnancies (∼−60% and ∼−73%, respectively, compared to WT x WT), meanwhile, no differences were detected for WT female fetuses (Figure 3F). There was a greater contribution of CI-associated leak respiration to total ETS capacity in the placental LZ of both sexes in the α/+ x WT group compared to the WT x WT group (Figure 3G). However, in the OXPHOS state, the contribution of CI to total ETS was increased only in the placental LZ of WT females, and not males and this was observed for both the WT x α/+ and α/+ x WT compared to WT x WT pregnancies (Figure 3H). There were no differences in oxygen flux associated with fatty acid oxidation and CIV activity for female or male WT fetuses between the three crosses (Appendix A). Taken together, these results indicate that placental LZ mitochondrial bioenergetics of both female and male WT fetuses is modulated by littermate and maternal p110α deficiency.

### 3.3. Littermate and/or Maternal p110α Deficiency Alters the Expression of Mitochondrial-Related Proteins in the Placental LZ of WT Fetuses in a Sex-Specific Manner

To provide information on the molecular changes mediating alterations in mitochondrial bioenergetics with littermate and maternal p110α deficiency, Western blotting was used to quantify the abundance of individual ETS complexes and additional mitochondrial-related proteins in the placental LZ of WT fetuses in our three pregnancy groups (WT x WT, WT x α/+, and α/+ x WT). There was a decreased abundance of CI and CII proteins in the placental LZ of WT females in both WT x α/+ and α/+ x WT pregnancies, compared to WT x WT (Figure 4A). In WT males, CI was instead significantly increased in both WT x α/+ and α/+ x WT pregnancies compared to WT x WT pregnancies and no difference in CII levels between the three pregnancy groups was found (Figure 4B). There was an increased abundance of CIII protein in WT females from the WT x α/+ compared to the WT x WT group, but again, no changes were seen in males (Figure 4A,B). CIV protein expression was decreased in the LZ of both WT female and male fetuses from α/+ x WT pregnancies when compared to WT x α/+. No differences were detected in CV protein abundance between the three groups regardless of fetal sex (Figure 4A,B). 

The abundance of citrate synthase, an indicator of mitochondrial density, and the abundance of PGC-1α, a transcription factor that promotes mitochondrial biogenesis, were altered in the LZ of male but not female WT fetuses, presenting with an increase for both proteins in WT x α/+ and α/+ x WT pregnancies, compared to WT x WT (Figure 4C,D). In contrast, the abundance of PPARγ, another transcription factor that regulates mitochondrial biogenesis, was reduced in the placental LZ of WT female fetuses of WT x α/+ pregnancies when compared with WT x WT, whilst no differences were observed for males (Figure 4E). Finally, we evaluated the abundance of UCP2, a protein that uncouples oxygen consumption from ATP synthesis, and found UCP2 was increased in the placental LZ of males, but not females, from WT x α/+ pregnancies, when compared to WT x WT (Figure 4F). Taken together, these data indicate that there are sex-dependent changes in the LZ abundance of key mitochondrial regulatory proteins in WT fetuses exposed to littermate and/or maternal p110α deficiency.

### 3.4. Littermate and/or Maternal p110α Deficiency Alters the Abundance of Key Growth and Metabolic Signalling Proteins in a Manner That Depends on Fetal Sex

To further understand the mechanisms underlying the sex-specific changes in placental LZ profile in WT fetuses exposed to littermate and maternal p110α deficiency, we evaluated the expression of key growth and metabolic signalling proteins in our three pregnancy groups (WT x WT, WT x α/+, and α/+ x WT: Figure 5A,B). The selected proteins are known to regulate cellular bioenergetics by linking endocrine signals with mitochondrial function [1]. In WT female fetuses, the LZ abundance of total AKT was not affected, but the level of active phosphorylated AKT was decreased in both WT x α/+ and α/+ x WT pregnancies compared to WT x WT (Figure 5C,D). Meanwhile, WT males showed an increased abundance of total AKT in addition to reduced AKT activation only in α/+ x WT pregnancies compared to WT x WT and/or WT x α/+ (Figure 5C,D). WT females in α/+ x WT pregnancies exhibited increased total MAPK 44/42 abundance compared to those from WT x WT and WT x α/+ pregnancies, but no change was seen in the level of active phosphorylated protein. Furthermore, no changes in total or activated MAPK 44/42 were seen for the placental LZ of WT males, with similar values observed for all three pregnancy groups. Total p38 MAPK was also greater in the placental LZ of WT females from α/+ x WT pregnancies, but phosphorylated active levels were reduced in both WT x α/+ and α/+ x WT pregnancies compared to WT x WT. In contrast, in males, only changes in p38 MAPK were observed between the WT x α/+ and α/+ x WT pregnancy groups, with a tendency for increased total p38 MAPK, and a significant decrease in activated p38 MAPK. Together, these data suggest that growth and metabolic signalling proteins in the WT placenta are affected by both littermate and maternal p110α deficiency. Moreover, the nature of these effects depends on fetal sex.

## 4. Discussion

Using genetic disruption of p110α signalling as a tool, we demonstrated the important influence of the maternal and intrauterine environments on feto-placental growth and placental mitochondrial respiration/metabolism. Specifically, by comparing to pregnancies where both the mother and all fetuses are WT, placental LZ respiratory capacity of WT fetuses of both sexes varied when either littermates and/or the mother was p110α deficient. Moreover, there were alterations in placental LZ mitochondrial bioenergetic capacity that were associated with sex-specific changes in the abundance of mitochondrial-related, growth, and metabolic proteins in the placenta of WTs exposed to littermate and/or maternal p110α deficiency. Together, these data may have relevance for understanding divergent fetal outcomes in pregnancies associated with different gestational conditions. They also support the concept of developing treatments that are targeted to the placenta, which are tailored to the sex of the fetus.

In previous studies done by our laboratory, we demonstrated that conceptus p110α deficiency leads to reduced fetal and placental growth [21,24,56]. Furthermore, α/+ dams have metabolic imbalances before and during pregnancy, which may relate to the altered fetal outcomes observed in this study and in our previous work [24,54,56]. In particular, compared to WT mothers carrying only WT fetuses in the litter, α/+ dams are lighter and unable to shift their glucose handling during pregnancy to favour fetal growth [54]. The novelty of the current work is the finding that WT males in a WT mother who carried α/+ fetuses (WT x α/+) are heavier than those from WT-only pregnancies (WT x WT). Moreover, this effect is lost if the pregnancy was carried by a mutant α/+ dam (α/+ x WT). These data are consistent with other work showing the important influence of genetically altered littermates on the intrauterine growth of their siblings [21,59]. This study therefore extends those findings to show that the influence of an altered intrauterine environment caused by sharing a litter with a mutant fetus (in this case α/+ fetus), is dependent on fetal sex. It is well established that males grow faster than females in utero [1,60,61,62,63]. In addition, recent data indicate that males are more growth-impaired than females in response to fetal p110α deficiency [56]. Therefore, one interpretation may be that WT males can grow more when there is less competition for maternally-supplied metabolic substrates from α/+ littermates. However, this innate ability of WT males to enhance their growth is hindered when the maternal ability to support the pregnancy is compromised by p110α deficiency.

We also found that mitochondrial oxygen consumption in the placental LZ of WT fetuses is responsive to the environment provided by the mother and/or littermates carrying the α/+ mutation. Overall, changes in LZ mitochondrial OXPHOS capacity for WT fetuses was similar for those gestated with p110α deficient littermates (WT x α/+) and/or p110α deficient mother (α/+ x WT), compared to WTs from entire WT pregnancies (WT x WT). This indicates that littermate, and largely not maternal, p110α deficiency impacts mitochondrial respiratory capacity in the placental LZ of WT fetuses. For instance, placental LZ CII_OXPHOS_ and total ETS were greater for WT fetuses from either WT x α/+ or α/+ x WT pregnancies compared to WT x WT. However, the effect of p110α deficiency to increase LZ CI_Leak_ flux control ratio (CI_Leak_/total ETS) was only significant for WT fetuses from α/+ x WT pregnancies compared to WT x WT (values for WT x α/+ pregnancies were intermediate between the two groups). Moreover, some sex-specific changes were found in the effect of littermate p110α deficiency. In particular, LZ CI_OXPHOS_ flux control (CI_OXPHOS_/total ETS) ratio was elevated only in WT females, and reserve capacity lower only for WT males, in WT x α/+ and α/+ x WT compared to WT x WT pregnancies. Earlier work has demonstrated that overall, both fetal and maternal p110α deficiency results in defective placental endocrine output, namely altered expression of placental lactogen genes [24]. Other work has also reported sex-dependent differences in placental capacity to produce placental lactogens, as well as sex steroids in response to genetic manipulations [64,65]. Placental hormones can exert local effects with resultant impacts on the placental support of fetal development [65,66]. Hence, changes in placental endocrine output may modulate feto-placental phenotype, including mitochondrial respiration of WT littermates, and how this may be modified by fetal sex requires further work. Female and male fetuses also vary in their circulating hormones [67,68], and this could serve as an additional mechanism behind sexually-dimorphic placental mitochondrial phenotypes as reported in several gestational pathologies [69,70,71,72,73,74,75].

Sex-specific differences in WT fetuses exposed to littermate and/or maternal p110α deficiency were related to molecular changes in the placental LZ. For instance, WT females presented a decreased CI and CII protein abundance in both WT x α/+ and α/+ x WT pregnancies, which may explain their reduced mitochondrial respiratory capacity. However, CIII abundance was also up-regulated for WT females from the WT x α/+ parental cross. In contrast, abundance of CI was elevated in the LZ of WT males of both WT x α/+ and α/+ x WT pregnancies and no differences were found for CII or CIII mitochondrial complexes, which could explain the changes in mitochondrial OXPHOS regardless of the parental cross. Interestingly, CIV was similarly affected in both WT females and males, presenting a decreased protein expression only when the mother was p110α deficient (α/+ x WT), when compared to pregnancies generated by a father who was p110α deficient (WT x α/+). Other studies conducted on placentas from adverse maternal gestational environments such as malnutrition, hypoxia, preeclampsia, obesity and metabolic diseases have also reported changes in the abundance of mitochondrial complexes, but few if at all have explored if changes are sex-specific [31,33,37,69,76,77,78,79,80,81,82]. Together these data highlight there is an important role for maternal and intrauterine p110α deficiency in modulating the expression of mitochondrial respiratory complexes by the WT male and female placental LZ.

There were also sex-specific changes in the levels of mitochondria-related proteins in the placental LZ of WT fetuses exposed to littermate and/or maternal p110α deficiency. For instance, abundance of citrate synthase and PGC1α were increased only in WT males from both WT x α/+ and α/+ x WT pregnancies, suggesting a compensatory elevation in mitochondrial density and biogenesis which is in line with higher CI levels compared to WT x WT pregnancies. Additionally, PPAR and uncoupling protein UCP2 protein abundance were altered in WT females and WT males, respectively and only in the group with mutated littermates (WT x α/+). The relevance of these changes though are unknown, especially given that mitochondrial fatty acid oxidation and oxygen consumption in LEAK state were not different for the placental LZ of females or males regardless of the parental cross (no differences between WT x WT, WT x α/+ and α/+ x WT). The abundance and activity of AKT, MAPK 44/42 and p38MAPK signalling proteins in the LZ of WT fetuses were also sex-specifically impacted by littermate and/or maternal p110α deficiency. However, unlike the changes in LZ respiratory capacity and mitochondria-related proteins, the impacts were more pronounced for WT fetuses if the mother plus littermates carried the p110α mutation (the α/+ x WT cross). Compared to both WT x WT and WT x α/+pregnancies, WT females presented increased levels of MAPK 44/42 and p38MAPK, whilst males showed elevated AKT levels. Interestingly, both WT females and males presented decreased levels of activated AKT (phosphorylation/total protein ratio) in α/+ x WT pregnancies, but for females this was also observed for the WT x α/+ cross. AKT signalling is important for placental growth and transport functional capacity [26,83] and recent work has shown that signalling via AKT in the placental LZ can vary in a sex-dependant manner [1,3]. The relevance of our current findings though is unclear, as placental LZ weight was unchanged, or even greater (for males in the α/+ x WT cross) in pregnancies where there was a deficiency in p110α (WT x α/+ or α/+ x WT). However, our data may have implications for previous work showing that WT placentas exposed to different p110α deficiencies vary in their glucose and amino acid transport *in vivo* [21]. Finally, activation of p38MAPK was decreased in the LZ of WT female fetuses exposed to both littermate and maternal p110α deficiency (i.e. WT x α/+ and α/+ x WT pregnancies compared to WT x WT), but increased specifically in the LZ of WT male fetuses in WT x α/+ compared to α/+ x WT. The significance of sex-specific changes in the WT placenta to littermate and maternal p110α deficiency is unclear.

While our study has multiple strengths, it also has certain limitations. For example, the evaluation of maternal food intake would be very helpful to understand how maternal metabolism and nutrient resources can impact on the changes observed in the placental LZ. Another limitation is that we only examined one gestational age and the changes observed in this study are likely to be a result of placental adaptations occurring earlier. Therefore, future experiments should evaluate additional gestational days to assess the ontogeny of sex-specific adaptations in placenta function. Lastly, our study was only focused on the LZ and we did not study the endocrine junctional zone. Therefore, future work should investigate the changes occurring in this placental region, which is also critical for the success of the pregnancy. Indeed, we have recently shown in mice that small for gestational age fetuses have changes in the function of the junctional zone [84]. Therefore, it is plausible that additional sex-specific adaptations are occurring in this placental region.

To conclude, our data emphasize the significance of the maternal and intrauterine environment in the control of fetal development. Furthermore, modulation of fetal development by the maternal and intrauterine environment is accompanied by changes in placental formation, energetic capacity and abundance of proteins regulating mitochondrial density, lipid metabolism, growth and nutrient handling of female and male fetuses. Our finding may have relevance for understanding the pathways leading to reduced fetal growth in suboptimal maternal environments, as well as for fetal outcomes in the context of multiple gestations/litter bearing species [3].

## Figures and Tables

**Figure 1 cells-12-00797-f001:**
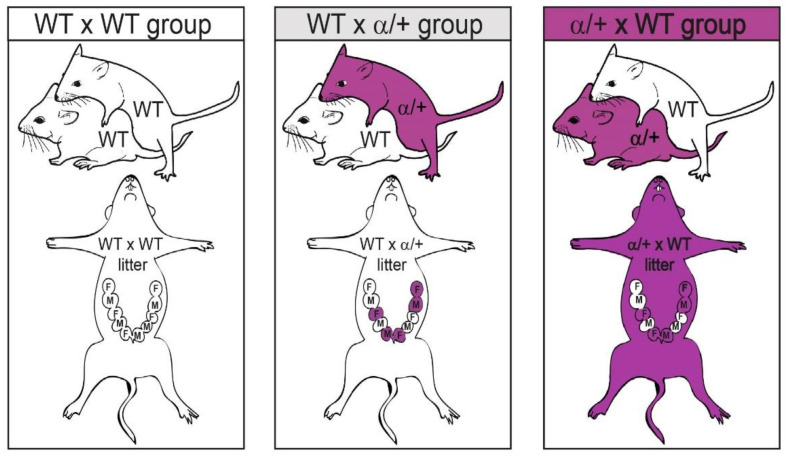
Illustrative figure representing the crosses used in the study. WT = wild-type; α/+ = Heterozygous *Pik3ca*-D933A mice; F = female fetus; M = male fetus.

**Figure 2 cells-12-00797-f002:**
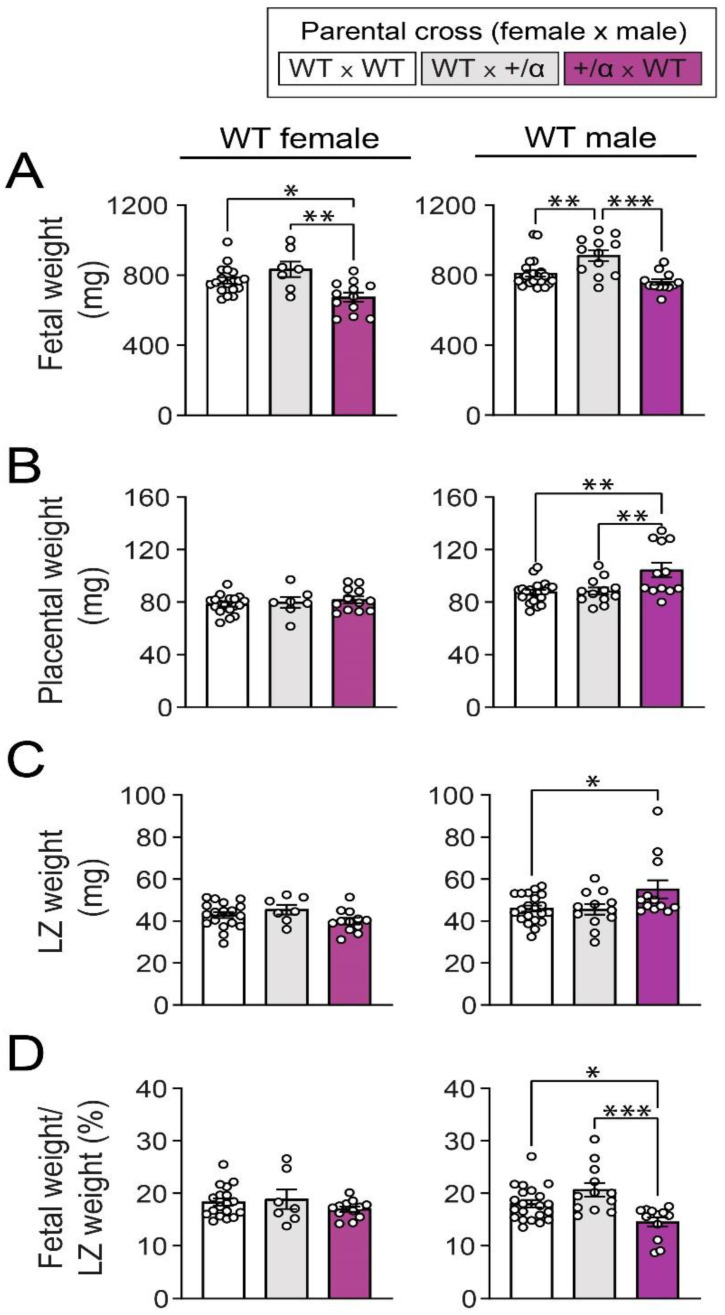
Fetal and placental growth of WT conceptuses in response to littermate and/or maternal p110α deficiency. Fetal weight (**A**), placenta weight (**B**), LZ weight (**C**), and fetal weight/LZ weight (**D**) in females and males on day 18 of pregnancy. Data are from WT fetuses generated by WT x WT, WT x α/+, and α/+ x WT parental crosses (*n* = 1–2 fetuses/sex/dam with 5–12 dams/group) and are displayed as individual data points with mean ± S.E.M. Data were analysed by one-way ANOVA with Tukey post hoc pairwise comparisons (* *p* < 0.05, ** *p* < 0.01, *** *p* < 0.001, pairwise comparison). LZ: labyrinthine zone.

**Figure 3 cells-12-00797-f003:**
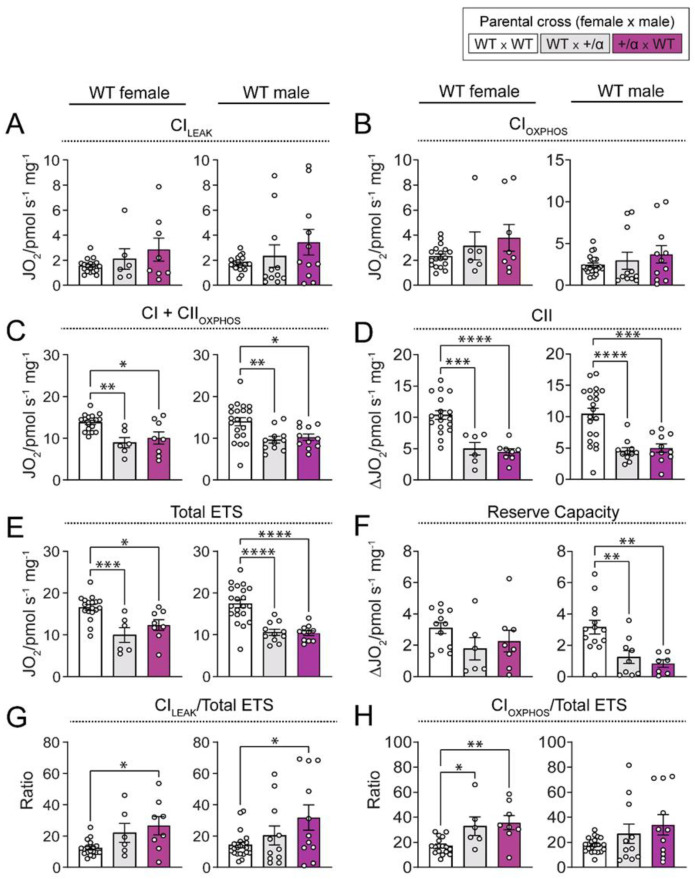
Placental mitochondrial bioenergetics of WT conceptuses in response to littermate and/or maternal p110α deficiency. Oxygen consumption in the placental LZ associated with CI_LEAK_ (**A**), CI_OXPHOS_ (**B**), CI + CII_OXPHOS_ (**C**), CII (**D**), total ETS (**E**), reserve capacity (**F**), CI_LEAK_/total ETS (**G**), and CI_OXPHOS_/total ETS (**H**) for females and males on day 18 of pregnancy. Data are from WT fetuses generated by WT x WT, WT x α/+, and α/+ x WT parental crosses (*n* = 1–2 fetuses/sex/dam with 5–12 dams/group) and are displayed as individual data points with mean ± S.E.M. Data were analysed by one-way ANOVA with Tukey post hoc pairwise comparisons (* *p* < 0.05, ** *p* < 0.01, *** *p* < 0.001, **** *p* < 0.0001, pairwise comparison).

**Figure 4 cells-12-00797-f004:**
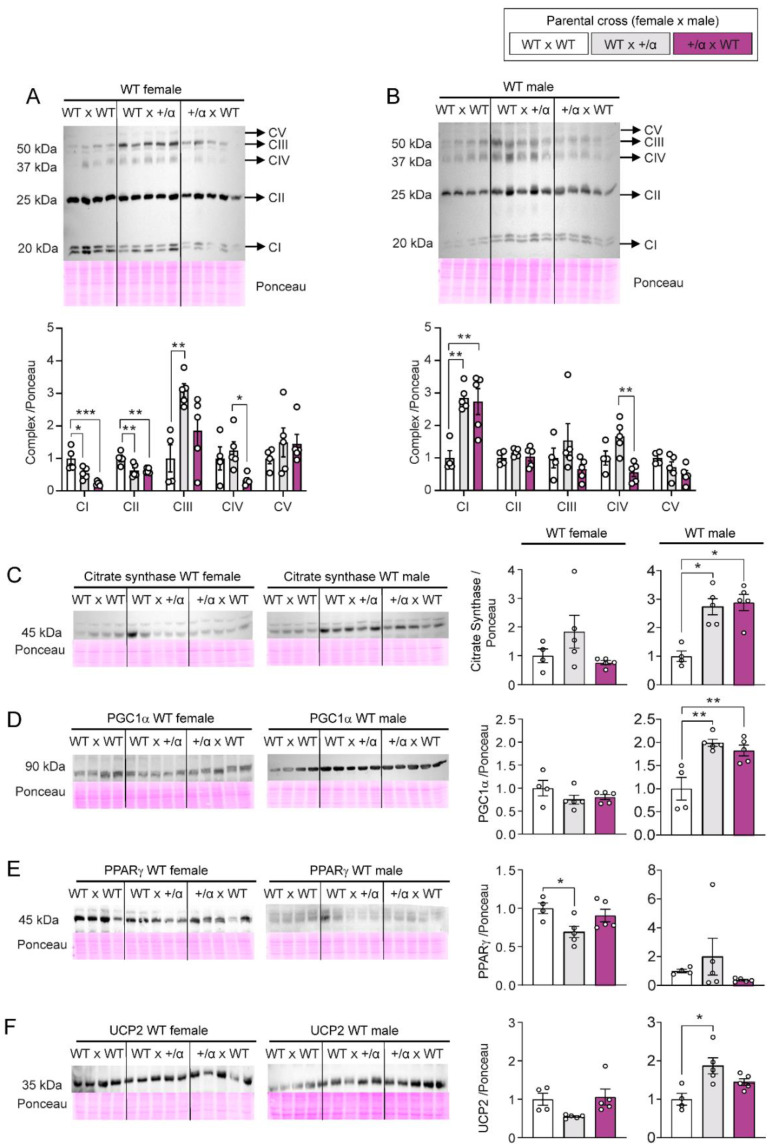
Protein abundance of mitochondrial complexes and key mitochondrial regulatory proteins in placental labyrinth of WT conceptuses in response to littermate and/or maternal p110α deficiency. Relative protein abundance of mitochondrial complexes in females (**A**) and males (**B**), as well as citrate synthase (**C**), PGC1α (**D**), PPARγ (**E**), and UCP2 (**F**) in females and males. Representative images from each antibody and Ponceau staining are included. Data are from 1 WT fetus per dam generated by WT x WT (*n* = 4), WT x α/+ (*n* = 5), and α/+ x WT (*n* = 5) parental crosses and are displayed as individual data points with mean ± S.E.M. Data were analysed by one-way ANOVA and Tukey post hoc pairwise comparisons (* *p* < 0.05, ** *p* < 0.01, *** *p* < 0.001, pairwise comparison).

**Figure 5 cells-12-00797-f005:**
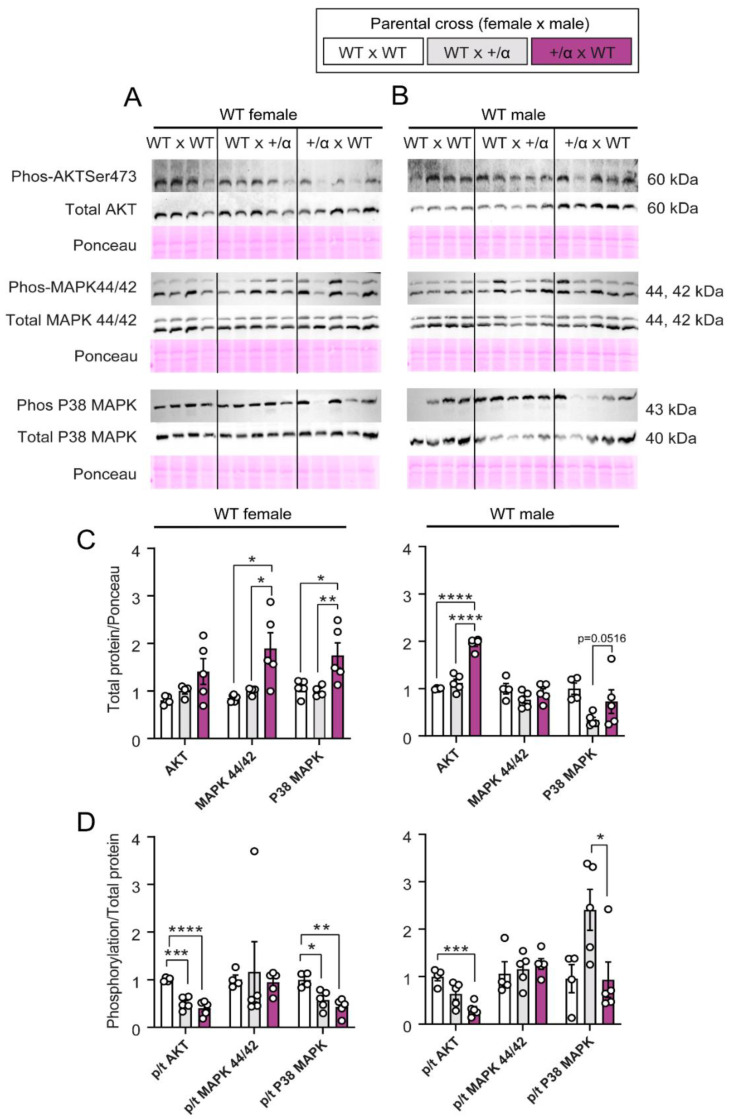
Abundance of growth and metabolic signalling proteins in placental labyrinth of WT conceptuses in response to littermate and/or maternal p110α deficiency. Female (**A**) and male (**B**) representative images from each antibody immunodetection and Ponceau staining for phosphorylated and total AKT, AMPKα, MAPK 44/42, and P38 MAPK. Total AKT, AMPKα, MAPK 44/42, and P38 MAPK protein levels in females and males (**C**), and AKT, AMPKα, MAPK 44/42, and P38 MAPK phosphorylation levels as a ratio to total protein in females and males (**D**). Data are from 1 WT fetus per dam generated by WT x WT (*n* = 4), WT x α/+ (*n* = 5), and α/+ x WT (*n* = 5) parental crosses and are displayed as individual data points with mean ± S.E.M. Data were analysed by one-way ANOVA with Tukey post hoc pairwise comparisons (* *p* < 0.05, ** *p* < 0.01, *** *p* < 0.001, **** *p* < 0.0001, pairwise comparison). *p* = phosphorylated, t = total.

## Data Availability

All relevant data are within the paper and available upon reasonable request.

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
