# Peer review of "Maternal and Intrauterine Influences on Feto-Placental Growth Are Accompanied by Sexually Dimorphic Changes in Placental Mitochondrial Respiration, and Metabolic Signalling Pathways"

_cells, 2023, doi:10.3390/cells12050797_

Round 1
Reviewer 1 Report
The manuscript is well-written and presents the data in a succinct manner that is easily interpreted. In addition, the authors should be commended for their restraint when interpreting the results and for not overstating relevance or implications while highlighting the novelty and importance of their work.
Small grammatical errors are present throughout Ln 77,203, 275,317,332
Given the centrality of the LZ to the manuscript, the introduction could benefit from a brief introduction of why it was chosen over the junctional zone and the similarities and/or differences between the human placenta and its role in pathology. While it is acknowledged that the authors have mentioned transport region in methods inclusion of additional information would increase the relevance of this data and the impact of the study to an unfamiliar reader.
Although the addition of Cyt C is mentioned in the manuscript methods, the reason why is absent. Assuming that this was to assess mitochondrial integrity post cryopreservation and saponin exposure. In addition, could the authors comment on why Cyt C was added at the end of the SUIT protocol instead of during OXPHOS state and if this may be a more or less valid measure of cyt c release and mitochondrial integrity.
Please expand on why data points are expressed individually “Ln159 when possible”. This is indirect and requires clarification.
Given the interesting differences and changes between the parental groups and fetal outcomes, was food consumption accounted for in the study analysis or collected and assessed for variability between the mothers. As food intake and availability in dams (as the authors allude to in the discussion) have also been attributed to fetal and metabolic changes in gestational models.
Could the authors comment on why the fetal weight ratio was derived using the LZ alone and not the total placental weight.
Author Response
- The manuscript is well-written and presents the data in a succinct manner that is easily interpreted. In addition, the authors should be commended for their restraint when interpreting the results and for not overstating relevance or implications while highlighting the novelty and importance of their work.
R: We want to thank the reviewer for their comments and feedback.
- Small grammatical errors are present throughout Ln 77,203, 275,317,332
R: Amended
- Given the centrality of the LZ to the manuscript, the introduction could benefit from a brief introduction of why it was chosen over the junctional zone and the similarities and/or differences between the human placenta and its role in pathology. While it is acknowledged that the authors have mentioned transport region in methods inclusion of additional information would increase the relevance of this data and the impact of the study to an unfamiliar reader.
R: Information was added in the introduction, now the text reads like this:
Text: “Mouse and human placentas are relatively similar in exchange, endocrine function and morphology, as both are haemochorial in nature. However, the murine placenta is divided in two distinguished regions, the junctional zone (responsible for endocrine production) and the labyrinth zone (LZ; substrate exchange) [13–15]. Whereas in the human, the villous syncytiotrophoblast performs both functions. Nevertheless, comparative proteomic and transcriptomic analyses of mouse placental LZ and human placental villous samples showed that over 80% of genes implicated in mouse placental phenotypes are co-expressed in both species [16].”
- Although the addition of Cyt C is mentioned in the manuscript methods, the reason why is absent. Assuming that this was to assess mitochondrial integrity post cryopreservation and saponin exposure. In addition, could the authors comment on why Cyt C was added at the end of the SUIT protocol instead of during OXPHOS state and if this may be a more or less valid measure of cyt c release and mitochondrial integrity.
R: Apologies for not including this information. We did add cytC to assess mitochondrial integrity and this was added during maximum OXPHOS capacity (after succinate to activate CI and CII pathways in the presence of ADP) protocol. See lines 143-146 In addition, a reference was added to validate the use of exogenous cytC. See line 148.
Text: “To provide information about outer membrane integrity, exogenous cytochrome c (10µM) was added during maximum OXPHOS capacity (after succinate addition to activate CI and CII pathways in the presence of ADP), and LZ samples showing a >30% increase in oxygen consumption were excluded (a total of 6 placentas from different groups were excluded from the study) as suggested by others [58].”
- Please expand on why data points are expressed individually “Ln159 when possible”. This is indirect and requires clarification.
R: We deleted “when possible” as all data points were shown in the figures
- Given the interesting differences and changes between the parental groups and fetal outcomes, was food consumption accounted for in the study analysis or collected and assessed for variability between the mothers. As food intake and availability in dams (as the authors allude to in the discussion) have also been attributed to fetal and metabolic changes in gestational models.
R: We thank the reviewer for their comment. Unfortunately, we did not measure food intake in these animals as they were housed in groups of 5 dams per cage. We have added a sentence in the manuscript highlighting the idea of the reviewer. See lines 433-433
Text: “While our study has multiple strengths, it also has certain limitations. For example, the evaluation of maternal food intake would be very helpful to understand how maternal metabolism and nutrient resources can impact on the changes observed in the placental LZ.”
- Could the authors comment on why the fetal weight ratio was derived using the LZ alone and not the total placental weight.
R: As this study was focused on the placental LZ, which is the key determinant for nutrient support to the fetus, we decided to express fetal weight as a proportion of LZ weight. We have calculated fetal weight as a proportion of whole placental weight. These data are shown below and largely reflect data when the LZ weight is used instead; major difference is that α/+ x WT ratios were statistically lower than WT x WT (-16.38%) and WT x α/+ (-22.24%)). If the reviewer would like, we can add these graphs to the paper. We have included a sentence describing our rationale for using the fetal weight to LZ weight ratio in the revised text. please see lines 197-201

Figure. Ratio between fetal weight and placental weight from female and males WT conceptuses in response to littermate and/or maternal p110α deficiency. Day 18 of pregnancy. Data are from WT fetuses generated by WT x WT, WT x α/+ and α/+ x WT parental crosses (n=1-2 fetuses/sex/dam with 5-12 dams/group) and are displayed as individual data points with mean ± S.E.M. Data were analyzed by one-way ANOVA with Tukey post hoc pairwise comparisons (*P<0.05, **P<0.01, ***P<0.001, pairwise comparison).
Text: “Since our study is focused on responses of the placental LZ, which determines nutrient supply to the fetus for growth, we calculated the ratio between fetal weight and placental LZ weight. This calculation revealed that fetal weight as a proportion of LZ weight was not different among the groups in females but was lower in males from the α/+ x WT group when compared to WT x WT and WT x α/+ groups (Figure 2D).”
Reviewer 2 Report
In the present study Salazar-Petres et al. have described how the maternal and the intrauterine environment can modulate fetal and placental growth by altering mitochondrial function as well as key metabolic signaling pathways. Moreover, the authors also highlight how these differences are dependent on fetal sex, a hot topic in placental research. Overall, this is a well-conducted and relevant study with compelling evidence supporting the authors’ claims. Thus, I consider that it should be accepted for publication in Cells. However, I have identified some minor issues that need to be amended prior to final acceptance.
Minor comments:
- A paragraph highlighting the limitations of the study and the future perspectives should be added to the discussion.
- The authors should add the molecular weight (kDa) of the proteins analyzed by Western blotting to the figures.
- Line 41: A full stop should be added after “[…] to the fetus [7-10]”
- Line 275 should read “[…] proteins in […]”
- Line 296 should read “Figure 5. Abundance of growth […]”
- Line 421 should read “Our findings […]”
Author Response
In the present study Salazar-Petres et al. have described how the maternal and the intrauterine environment can modulate fetal and placental growth by altering mitochondrial function as well as key metabolic signaling pathways. Moreover, the authors also highlight how these differences are dependent on fetal sex, a hot topic in placental research. Overall, this is a well-conducted and relevant study with compelling evidence supporting the authors’ claims. Thus, I consider that it should be accepted for publication in Cells. However, I have identified some minor issues that need to be amended prior to final acceptance.
- A paragraph highlighting the limitations of the study and the future perspectives should be added to the discussion.
R: A paragraph was included at lines 430-441
“While our study has multiple strengths, it also has certain limitations. For example, the evaluation of maternal food intake would be very helpful to understand how maternal metabolism and nutrient resources can impact on the changes observed in the placental LZ. Another limitation is that we only examined one gestational age and the changes observed in this study are likely to be a result of placental adaptations occurring earlier. Therefore, future experiments should evaluate additional gestational days to assess the ontogeny of sex-specific adaptations in placenta function. Lastly, our study was only focused on the LZ and we did not study the endocrine junctional zone. Therefore, future work should investigate the changes occurring in this placental region, which is also critical for the success of the pregnancy. Indeed, we have recently shown in mice that small for gestational age fetuses have changes in the function of the junctional zone [86]. Therefore, it is plausible that additional sex-specific adaptations are occurring in this placental region."
- The authors should add the molecular weight (kDa) of the proteins analyzed by Western blotting to the figures.
R: As requested, figures have been updated
- Line 41: A full stop should be added after “[…] to the fetus [7-10]”
R: Amended
- Line 275 should read “[…] proteins in […]”
R: Amended
- Line 296 should read “Figure 5. Abundance of growth […]”
R: Amended
- Line 421 should read “Our findings […]”
R: Amended